# GSRec: A Graph-Sequence Recommendation System Based on Reverse-Order Graph and User Embedding

**Xulin Ma** [1], **Jiajia Tan** [1], **Linan Zhu** [1], **Xiaoran Yan** [2] and **Xiangjie Kong** [1,*]

1    College of Computer Science and Technology, Zhejiang University of Technology, Hangzhou 310023, China; 2112112045@zjut.edu.cn (X.M.); 2112112058@zjut.edu.cn (J.T.); zln@zjut.edu.cn (L.Z.)
2    Research Institute of Artificial Intelligence, Zhejiang Lab, Hangzhou 310023, China; yanxr@zhejianglab.com
*    Correspondence: xjkong@zjut.edu.cn

**Abstract:** At present, sequence-based models have various applications in recommendation systems; these models recommend the interested items of the user according to the user's behavioral sequence. However, sequence-based models have a limitation of length. When the length of the user's behavioral sequence exceeds the limitation of the model, the model cannot take advantage of the complete behavioral sequence of the user and cannot know the user's holistic interests. The accuracy of the model then goes down. Meanwhile, sequence-based models only pay attention to the sequential signals of the data but do not pay attention to the spatial signals of the data, which will also affect the model's accuracy. This paper proposes a graph sequence-based model called GSRec that combines Graph Convolutional Network (GCN) and Transformer to solve these problems. In the GCN part we designed a reverse-order graph, and in the Transformer part we introduced the user embedding. The reverse-order graph and the user embedding can make the combination of GCN and Transformer more efficient. Experiments on six datasets show that GSRec outperforms the current state-of-the-art (SOTA) models.

**Keywords:** graph neural network; sequential recommendation; representation learning

## 1. Introduction

In the past few decades, because of the rapid development of the Internet, individuals can collect various information simply. However, massive amounts of information often make individuals unable to find items they are interested in; as a consequence, the personalized recommendation system is proposed. The personalized recommendation system can provide information filtering for users and screen out items that users need.

Many experiments [1–5] show that the user's current interest is dynamic in nature, and the user's behavioral sequence influences this interest. For example, if a user buys a phone, he is more likely to buy a phone shell, even though he would not usually buy one. In order to capture this dynamic interest, many models [6–8] based on users' behavioral sequences have been proposed. These models learn the sequential signals between items and predict which item is most likely to be purchased based on the user's behavioral sequence.

Although sequence-based models have achieved great success in recommendation systems, sequence-based models have the following disadvantages. First, the length of the sequence is limited. When the length of the user's behavioral sequence exceeds the limit of the model, the model cannot fully capture the user's interest, thus reducing the model's accuracy. Furthermore, if the length is too long, it will inevitably increase the model prediction time. For example, in a scenario where the sequence length is 5, and the model predicts that the user is likely to buy a computer based on the last five items purchased by the user, due to the length limit of the model, the model cannot observe the complete purchase sequence of the user and, therefore, cannot find that the item had been

purchased long ago. Second, sequence-based models only pay attention to the sequential signals between items but do not pay attention to the spatial signals of items and users, which will also affect the effectiveness of the model. This is because, without the spatial signals, the model cannot learn a user's interest from other users.

Overall, the sequence-based models have the following issues: (1) Due to the limitation of sequence length, the sequence-based models may discard the long-term behavior of users and cannot fully capture the user's interest. (2) Sequence-based models focus on the user's recent behavior and ignore their complete behavior. (3) Sequence-based models only focus on the user's behavior sequence and ignore the spatial signal between users and items.

To solve the above questions, we propose a novel model called GSRec that combines Graph Convolutional Network (GCN) and Transformer. We use GCN because GCN constructs a graph based on the user's historical behavior and the nodes in the graph aggregate information about their neighbors. This means that GCN can learn the user's holistic interest and spatial signals. At the same time, in order to ensure a GCN can better learn the user's holistic interest, and thus make the combination of GCN and Transformer more efficient, we design a reverse-order graph in the GCN part. The contributions of this work are summarized as follows:

- We present a graph-sequence model that can fully use the users' complete behavior sequence and the spatial signals of data to increase data utilization and improve the accuracy of recommendations.
- To make the combination of GCN and Transformer more efficient, we design the reverse-order graph in GCN and the user embedding in Transformer.
- We conducted experiments on six datasets, and the results show that our model outperforms the current state-of-the-art (SOTA) models.

## 2. Related Work

Our work is related to three lines of research: sequence-based recommendation, GCN-based recommendation, and hybrid recommendation. We review the recent advances in these areas in the following sections.

### 2.1. Sequence-Based Recommendation

The task of sequence-based recommendation is to predict the next item the user will buy based on the user's behavioral sequence. The Markov chain-based model is a classic sequence-based recommendation model [9,10]. Because of the development of deep learning, RNN-based models were introduced into sequence-based recommendations. Bal et al. used the RNN-based model GRU4Rec [11] for the first time to predict the next item that the user might be interested in. However, because Markov assumes that the current interaction only depends on one or several recent interactions, the results predicted by models would only be dependent on the user's most recent behavioral sequence. In addition, CNN-based models were also introduced into sequential recommendations, such as Caser [12]. Since CNN-based models do not have a strong ordering assumption for the interaction in the sequence, the CNN-based recommendations can make up for the disadvantage based on RNN to a certain extent. Transformer [13], as a sequence-based model, achieved state-of-the-art performance and efficiency for machine translation tasks because of 'self-attention'. Therefore, an attention mechanism was introduced into sequence-based recommendations [14], and Wang et al. [15] proposed SASRec, which focuses more on the whole sequence instead of the most recent behavioral sequence. In addition, Li et al. [16] proposed TiSASRec, which is based on SASRec and uses the temporal factor, and Sun et al. [7] proposed Bert4Rec using Bidirectional Encoder Representations from Transformers (BERT). Yukuo et al. [17] introduced multiple interests into sequence-based recommendations to improve the model's performance. Although the above models are efficient, they require a fixed length, which means that if the user's sequence exceeds this length these models will not be able to express the user's interests fully. In order to

solve this problem, Bai et al. proposed LSDA [18], which uses multiple LSTM to learn the whole sequences of users, and Bo et al. proposed HAM [19]; this model uses MLP to learn the whole sequences of users. Nevertheless, these models only pay attention to the sequence and do not pay attention to the spatial signals of data.

*2.2. GCN-Based Recommendation*

Initially, traditional matrix factorization models [20,21] were used for the recommendation system. With the rise in deep learning methods, traditional matrix factorization models have been gradually replaced by deep neural networks, especially graph neural networks (GNN). Berg et al. [22] utilized a graph convolutional neural network in a recommendation system, and then Wang et al. [23] improved it and proposed NGCF. However, these models use the Laplacian matrix, and the matrix operation is troublesome; hence, He et al. [24] simplified the matrix operation and proposed LightGCN, which improved the efficiency of the operation in sparse datasets. Fan et al. [25] improved LightGCN by changing the global graph into a subgraph to improve the generalization of the model. In addition, Liang et al. [26] used auto-encoders for recommendations, which can effectively provide feedback on the implicit data of users. At the same time, Wang et al. [27] combined the attention mechanism with the graph neural network, proposed GAT, and applied it to recommendation systems [28]. Some scholars also use dynamic graphs to improve the recommendation ability of models [29–32]. GCN-based models can fully use the user's behaviors and can also use the spatial signal of the data to learn the user's holistic interests. However, GCN-based models cannot utilize the user's temporal signals and capture the recent interests effectively.

*2.3. Hybrid Recommendation*

Since graph convolutional neural networks can handle graph-structured data well, some scholars combine GNN with other models and apply them to sequence-based recommendation systems. For example, SR-GNN [33], MA-GNN [34], and DGCN [35] combine GNN and GRU, which improves the generalization ability of the model. APGNN [36] combines GNN and attention mechanisms. However, these models that use GRU will focus too much on the recent behavior of users and ignore their complete behavior, and GRU's computational complexity is high. Some scholars have also introduced a memory network into the recommendation system; thereby, the matrix can more explicitly and dynamically store and update the historical interactions in the sequence to improve the expressive ability of the model. For example, Chen et al. [37] and Huang et al. [38] added a memory network to the GRU to improve the expression ability of the model. Yuan et al. [39] used a memory network for session-based recommendations, and Wang et al. [40] used an attention network for the next location recommendation. Tan et al. [41] and Hsu et al. [42] used an attention graph for sequential recommendations. Meanwhile, some scholars introduce models that pay more attention to sequence. Zhu et al. proposed GES [43], combining GNN and Transformer, but this model focuses too much on the user's recent behavioral sequence. These hybrid models combine GCN-based models and sequence-based models, but the above models still focus on the recent behavior sequence of users and lack attention to the complete behavior of users.

**3. Preliminaries**

Researchers created the GCN to extract features from graph-structured data. Let a graph $\mathcal{G} = (\mathcal{V}, \mathcal{E})$ with node $v \in \mathcal{V}$, edge $(v, v') \in \mathcal{E}$. $\mathbf{H}^0 \in \mathbb{R}^{n \times d}$ is the original node embedding matrix, $n$ is the number of nodes, and $d$ is the embedding dimension of a node. $\mathbf{H}^l \in \mathbb{R}^{n \times d}$ is $l$-th layer hidden state of nodes. The original GCN [44] model follows the layer-wise propagation rule:

$$\mathbf{L} = \mathbf{D}^{-\frac{1}{2}} \tilde{\mathbf{A}} \mathbf{D}^{-\frac{1}{2}} \tag{1}$$

$$\mathbf{H}^{(l+1)} = \sigma\left(\mathbf{L}\mathbf{H}^{(l)}\mathbf{W}^{(l)}\right) \tag{2}$$

where $\tilde{\mathbf{A}} = \mathbf{A} + \mathbf{I}$ is the added self-connections adjacency matrix of graph $\mathcal{G}$. $\mathbf{I} \in \mathbb{R}^{(n_u+n_i)\times(n_u+n_i)}$ is the identity matrix. $\mathbf{D}$ denotes the degree matrix and $\mathbf{W}^{(l)}$ is the $l$-th layer trainable weight matrix. $\sigma(\cdot)$ denotes an activation function. As summarized in [44], during GCN training the updating process of each node follows two steps, aggregation and combination, which are defined as

$$\mathbf{h}_{agg}^{(l)} = \sigma\left(\mathbf{W}^{(l)} \cdot \mathrm{AGG}\left(\left\{\mathbf{h}_{v'}^{(l-1)}, \forall v' \in A(v)\right\}\right)\right) \tag{3}$$

$$\mathbf{h}_v^{(l)} = \mathrm{COMBINE}\left(\mathbf{h}_v^{(l-1)}, \mathbf{h}_{agg}^{(l)}\right) \tag{4}$$

where $A(v)$ is the set of adjacent nodes $v$ and $\mathrm{AGG}(\cdot)$ is an aggregation function aggregating hidden features from neighbor nodes $v$. Some aggregation functions have been studied, such as mean-pooling, max-pooling [45], and attention mechanism [27]. $\mathbf{W}^{(l)}$ is the $l$-th layer trainable weight matrix. $\mathrm{AGG}(\cdot)$ denotes aggregated neighbors' embeddings of node $v$ at $l$-th layer. $\mathrm{COMBINE}(\cdot)$ is a combination function that combines node $v$ self-embedding and aggregated neighbors' embeddings, whose optional operators include element-wise product, concatenation [45], and so on. In the original GCN, there is no explicit combination step because the adjacency matrix in the original GCN has self-connections. Hence, in the aggregation step the node self-embedding has been combined with its neighbors' features.

## 4. Methodology

In this section, we will introduce our GSRec model in detail and the summary of key notations is shown in Table 1.

**Table 1.** Summary of key notations.

| Key | Description |
| --- | --- |
| $n_u$ | the number of users |
| $n_i$ | the number of items |
| $U$ | the set of users, $U = \{u_1, u_2, \cdots, u_n\}$ |
| $I$ | the set of items, $I = \{i_1, i_2, \cdots, i_m\}$ |
| $\mathbf{I}$ | the identity matrix, $\mathbf{I} \in \mathbb{R}^{(n_u+n_i)\times(n_u+n_i)}$ |
| $L$ | the length of behavioral sequence |
| $S_u$ | the historical behavioral sequence of the user $u \in U$ |
| $s_l$ | the length of training sequence |
| $\mathbf{A}$ | adjacency matrix that has interactive information between users and items |
| $\mathbf{D}$ | degree matrix, $\mathbf{D} \in \mathbb{R}^{(n+m)\times(n+m)}$ |
| $d$ | the embedding size |
| $\mathbf{e}_u$ | an embedding of user $u$, $\mathbf{e}_u \in \mathbb{R}$ |
| $\mathbf{e}_i$ | an embedding of item $i$, $\mathbf{e}_i \in \mathbb{R}$ |
| $\mathbf{E}$ | an embedding matrix, $\mathbf{E} = [\mathbf{e}_{u1}, \cdots, \mathbf{e}_{un}, \mathbf{e}_{i1}, \cdots, \mathbf{e}_{im}]^T$ |
| $\mathbf{P}$ | position embedding, $\mathbf{P} \in \mathbb{R}^{s_l \times d}$ |
| $\theta$ | incentive factor in adjacency matrix, $\theta = 1/avglength$ |
| $index$ | the distance between an item and the last item the user clicked or purchased |

GSRec is devised to predict top-N ranked items with which the user will likely interact by exploiting existing user–item interaction information. As demonstrated in Figure 1, GSRec consists of two parts: The GCN layer and the sequence coding layer. In the GCN layer, the model first generates embeddings for each user and item. The model then generates the reverse-order graph through user–item interaction. Finally, the model uses GCN to extract high-dimensional spatial signals and learn the user's holistic interests, and the sequence coding layer uses multiple Transformer blocks to capture the users' sequential signals.

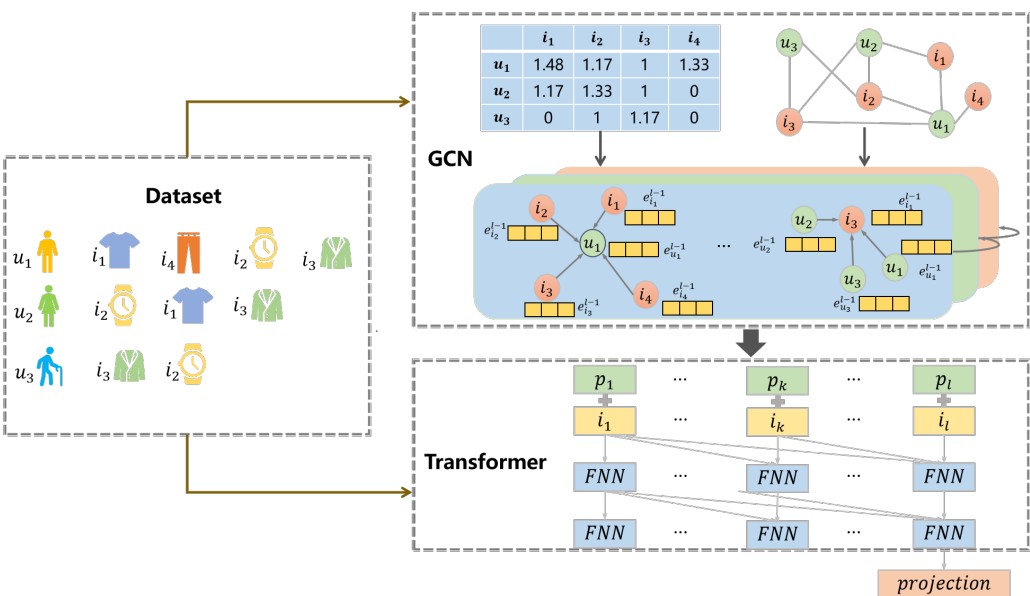

**Figure 1.** The architecture of the GSRec.

### 4.1. Problem Statement

In sequence-based recommendation, $U = \{u_1, u_2, \cdots, u_n\}$ is the set of users, $I = \{i_1, i_2, \cdots, i_m\}$ is the set of items, and $S_u = [i_1^u, \cdots, i_t^u, \cdots, i_L^u]$ represents the historical behavioral sequence of the user $u \in U$, where $i_t^u \in I$, $L$ is the length of interaction sequence for user $u$. Given a sequence of historical items, the probability that the user will interact with the item at the next moment $L + 1$: $p(i_{L+1}^u = i \mid S_u)$ is predicted, and the top-N items can be recommended to user $u$ according to the probabilities in descending order.

### 4.2. GCN Components

**Embedding Layer:** We can describe a user $u$ (an item $i$) with an embedding vector $\mathbf{e}_u \in \mathbb{R}^d$, $\mathbf{e}_i \in \mathbb{R}^d$, where $d$ denotes the embedding size. Therefore, we can build an embedding matrix $\mathbf{E}$:

$$\mathbf{E} = \begin{bmatrix} \mathbf{E}_U \\ \mathbf{E}_I \end{bmatrix} \quad \mathbf{E}_U = [\mathbf{e}_{u1}, \cdots, \mathbf{e}_{un}]^T \quad \mathbf{E}_I = [\mathbf{e}_{i1}, \cdots, \mathbf{e}_{im}]^T \tag{5}$$

where $n$ is the number of users and $m$ is the number of items. It should be noted that our task is to learn this embedding matrix; we update the embeddings by propagating them on the user–item interaction graph and finally predict the next item that the user will purchase or click according to this embedding matrix.

**Reverse-Order Graph:** In the traditional GCN-based model, the adjacency matrix is constructed from the user–item interaction graph:

$$\begin{cases} a(u_i, i_j) = 1 & u_i \text{ interact } i_j, \\ a(u_i, i_j) = 0 & \text{other.} \end{cases} \tag{6}$$

The behavioral information is shown in Figure 2a. This method treats all interactions as equally important, which means that all the interactions have the same status. However, Transformer pays more attention to the user's recent behavioral sequence; if we use this method to construct the adjacency matrix, the model will focus on the recent behavioral sequence of users. On the contrary, we want the model to treat holistic behaviors and recent behaviors synthetically; thus, we introduce the incentive factor $\theta$ to the adjacency matrix in

sparse and sequential datasets. In the following, *index* means the distance between an item and the last item the user clicked or purchased.

$$
\begin{cases}
a(u_i, i_j) = 1 + (\theta \times index)/2 & u_i \text{ interact } i_j. \\
a(u_i, i_j) = 0 & \text{other.}
\end{cases}
\tag{7}
$$

This method gives greater weight to the items that the user interacted with earlier, allowing GCN to focus more on these items. Through the above method, we can obtain the adjacency matrix $\mathbf{A}_{fus} \in \mathbb{R}^{n_u \times n_i}$ based on the incentive factor, and we can then obtain our reverse-order graph $\mathbf{L_r}$.

$$
\mathbf{L_r} = \mathbf{D}^{-\frac{1}{2}} \left( \begin{bmatrix} \mathbf{0}_1 & \mathbf{A}_{fus} \\ \mathbf{A}_{fus}^T & \mathbf{0}_2 \end{bmatrix} + \mathbf{I} \right) \mathbf{D}^{-\frac{1}{2}}
\tag{8}
$$

$\mathbf{I} \in \mathbb{R}^{(n_u + n_i) \times (n_u + n_i)}$ is the identity matrix and $\mathbf{D} \in \mathbb{R}^{(n_u + n_i) \times (n_u + n_i)}$ is the degree matrix. $n_u$ is the number of users and $n_i$ is the number of items. $\mathbf{0}_1 \in \mathbb{R}^{n_u \times n_u}$ and $\mathbf{0}_2 \in \mathbb{R}^{n_i \times n_i}$ are the null matrices. Here are the reasons why we do not use the incentive factor in dense or non-sequential datasets. In non-sequential datasets, we do not know users' behavioral sequences; thus, we cannot distinguish which behaviors are recent and which behaviors are from earlier. In dense datasets, the GCN module can learn the holistic behaviors of users better than the recent behaviors of users.

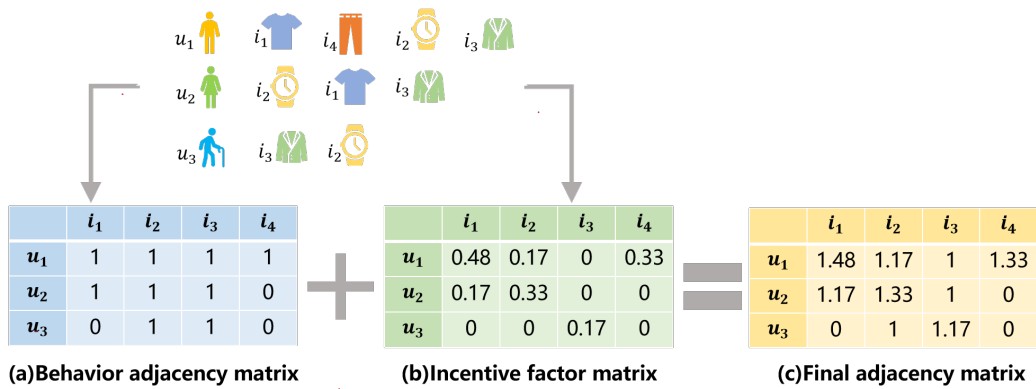

**Figure 2.** The process of the adjacency matrix.

**Graph Convolution:** Graph convolution is an important part of a recommendation system based on GCN. Its function is to learn the characteristics of nodes. The method of graph convolution can be expressed as

$$
\mathbf{e}_u^{(k+1)} = \text{AGG}(\mathbf{e}_u^{(k)}, \{ \mathbf{e}_i^{(k)} : i \in N_u \})
\tag{9}
$$

where AGG is a propagation function, which is the core of graph convolution and represents the representation of the next layer of the matrix. Many scholars have studied AGG [46,47], and although these methods have good results in graph classification they may be redundant for graph-based recommendations. Because, in graph-based recommendations, the initial embedding settings are random without valid information, the operation of graph convolution in GSRec is defined as

$$
\mathbf{e}_u^{(k+1)} = \sum_{i \in N_u} \frac{1}{\sqrt{|N_u|} \sqrt{|N_i|}} \mathbf{e}_i^{(k)}
\tag{10}
$$

$$
\mathbf{e}_i^{(k+1)} = \sum_{u \in N_i} \frac{1}{\sqrt{|N_u|} \sqrt{|N_i|}} \mathbf{e}_u^{(k)}
\tag{11}
$$

where $\frac{1}{\sqrt{|N_u|}\sqrt{|N_i|}}$ can avoid the scale of embeddings increasing with graph convolutional operations. After the propagation of $K$ layers, we need to combine each layer to obtain the final embedding representation. The combining rules are as follows:

$$\mathbf{e}_u = \sum_{k=0}^{K} \frac{1}{1+k} \mathbf{e}_u^{(k)} \tag{12}$$

$$\mathbf{e}_i = \sum_{k=0}^{K} \frac{1}{1+k} \mathbf{e}_i^{(k)} \tag{13}$$

If each element is processed more time complexity is required; thus, the matrix representation is given here:

$$\mathbf{E}^{(i)} = \mathbf{L_r}\mathbf{E}^{(i-1)} \tag{14}$$

*4.3. Sequential Encoder with Transformer*

After the graph convolutional network training, the embedding $\mathbf{E}$ aggregated the spatial signals and holistic interests of users. To capture the sequential signals, we used Transformer as an encoder.

**Transform the Sequence:** In different situations, we transform the sequence differently. If the count of users is less than the count of items in a dataset, we transform the sequence $S_u$ into a fixed-length sequence, and the index of a user is then added to the first of the sequence $s = (u_i, i_1, i_2, \cdots, i_L)$. If the count of users is more than or equal to the count of items in a dataset, we transform the sequence $S_u$ into a fixed-length sequence $s = (i_1, i_2, \cdots, i_L)$. Here are the reasons: When the count of users is less than the count of items, the users' embedding has more spatial signals, which makes the model more balanced. When the count of users is more than or equal to the count of items, the users' embedding has fewer spatial signals, which makes the model more unbalanced. After this, if the length of a sequence is more than or equal to $s_l$, we choose the most recent $s_l$ items, and if the length of a sequence is less than $s_l$, we add a 'padding' item to the left repeatedly until the length is equal to $s_l$. A random vector is used as the embedding for the padding item.

**Positional Embedding:** In Transformer, the self-attention model does not include any positive modules; in order to capture the position information of items we inject a learnable position embedding $\mathbf{P} \in \mathbb{R}^{(s_l+1)\times d}$ into the sequential embedding $\mathbf{E}_{seq} \in \mathbb{R}^{(s_l+1)\times d}$, and $s_l$ means the length of the sequence.

$$\tilde{\mathbf{E}} = \mathbf{E}_{seq} + \mathbf{P} = \begin{bmatrix} \mathbf{e}_u + \mathbf{p}_1 \\ \cdots \\ \mathbf{e}_{i_{s_l}} + \mathbf{p}_{s_l} \end{bmatrix} \tag{15}$$

**Self-Attention:** The attention mechanism mainly captures the correlation between representation pairs in the sequence model. An attention function can be described as mapping a query and the set of key-value pairs to an output, where the query, keys, values, and output are all vectors. The output is computed as a weighted sum of the values. The weight assigned to each value is computed by a compatibility function of the query with the corresponding key [13]. In Transformer, the attention function is

$$\text{Attention}\,(\mathbf{Q}, \mathbf{K}, \mathbf{V}) = \text{softmax}\left(\frac{\mathbf{Q}\mathbf{K}^\top}{\sqrt{d/h}}\right)\mathbf{V} \tag{16}$$

Multi-headed attention enables the model to pay joint attention to the information of different representation subspaces at different positions.

$$S = \text{MultiHead}-\text{SA}(\tilde{\mathbf{E}}) = \text{Concat}\left(\mathbf{h}_1, \ldots, \mathbf{h}_h\right)\tilde{\mathbf{E}}$$
$$\mathbf{h}_i = \text{Attention}\left(\mathbf{Q}\mathbf{W}_i^Q, \mathbf{K}\mathbf{W}_i^K, \mathbf{V}\mathbf{W}_i^V\right) \tag{17}$$

$\mathbf{W}_i^Q \in \mathbb{R}^{d \times d/h} \quad \mathbf{W}_i^K \in \mathbb{R}^{d \times d/h} \quad \mathbf{W}_i^V \in \mathbb{R}^{d \times d/h}$ are learnable parameters.

**Position-Wise Feed-Forward Network:** Self-attention is mainly based on linear projection. To construct a model with nonlinearity and interactions between different dimensions, we apply a *Position-wise Feed-Forward Network* to the outputs of the self-attention sublayer, which is applied to each position separately and identically. This consists of two linear transformations with a ReLU activation in between.

$$\mathbf{F}_i = \text{FFN}(\mathbf{S}_i) = \text{ReLU}\left(\mathbf{S}_i\mathbf{W}^{(1)} + b^{(1)}\right)\mathbf{W}^{(2)} + b^{(2)} \tag{18}$$

$\mathbf{W}^{(1)} \in \mathbb{R}^{d \times d}$, $\mathbf{W}^{(2)} \in \mathbb{R}^{d \times d}$, bias $b^{(1)}$, and bias $b^{(2)}$ are learnable parameters.

**Stacking Self-Attention Blocks:** $\mathbf{F}_i$ has aggregated previous embeddings of items through a self-attention block. To capture more complex item transition information, we use stacked self-attention blocks, and the *b*-th ($b > 1$) block is defined as

$$\mathbf{S}^{(b)} = \text{SA}\left(\mathbf{F}^{(b-1)}\right)$$
$$\mathbf{F}_{\mathbf{i}}^{(\mathbf{b})} = \text{FFN}\left(\mathbf{S}_i^{(b)}\right) \quad \forall i \in \{1, 2, \ldots, n\}, \tag{19}$$

The 1-st block is defined as $\mathbf{S}^{(1)} = \mathbf{S}$, $\mathbf{F}^{(1)} = \mathbf{F}$. In order to alleviate the problem of depth model overfitting and model instability, we perform the following operations:

$$g(x) = x + \text{Dropout}(g(\text{LayerNorm}(x))) \tag{20}$$

$$\text{LayerNorm}(x) = \alpha \odot \frac{x - \mu}{\sqrt{\sigma^2 + \epsilon}} + \beta \tag{21}$$

where $g(x)$ represents the self-attention layer or the feed-forward network, $\odot$ is an element-wise product, $\mu$ and $\sigma$ are the mean and variance of $x$, and $\alpha$ and $\beta$ are learned scaling factors and bias terms.

**Prediction Layer:** Through self-attention blocks, the model can extract information about consumed items, and we predict the next item based on $\mathbf{F}^{(b)}$. Finally, the model predicts item *i* score through the MF layer:

$$r_{i,t} = \mathbf{F}_t^{(b)}\mathbf{E}_I^T \tag{22}$$

where $r_{i,t}$ is the score of item *i* that the model predicts the user will purchase next, $\mathbf{E}_I$ is item embeddings, and T means transposition. We can generate recommendations by ranking the scores.

*4.4. Learning*

We exploit binary cross entropy (BCE) loss as the objective function; the BCE loss function is used to measure the difference between predicted values and real values:

$$loss = -\sum_{\mathcal{S}^u \in \mathcal{S}} \sum_{t=1}^{n} \left[\log(\sigma(r_{o_t,t})) + \sum_{j \notin \mathcal{S}^u} \log\left(1 - \sigma(r_{j,t})\right)\right] \tag{23}$$

The entire framework can be trained effectively by using end-to-end paradigm reverse propagation. The optimizer we used is Adaptive Moment Estimation (Adam) [48]. The Adam optimizer is an adaptive optimizer that combines the RMSProp optimizer and the

Momentum optimizer, which can adjust the learning rate based on historical gradient information:

$$\Delta w_t = \alpha \frac{m_t}{\sqrt{V_t + \epsilon}}$$
$$m_t = \beta_1 * m_{t-1} + (1 - \beta_1) * g_t \tag{24}$$
$$V_t = \beta_2 * V_{t-1} + (1 - \beta_2) * g_t^2$$

where $\alpha$ is the learning rate, $m_t$ is the momentum of the current step, $V_t$ is the variance of the current step, $\epsilon$ is a coefficient that increases the stability of the denominator, $\beta_1$ is the historical momentum retention rate, $\beta_2$ is the historical variance retention rate, and $g_t$ is the gradient. Additionally, to prevent overfitting a dropout strategy [49] is used for the linear layer of the model. The pseudocode for GSRec is shown in Algorithm 1.

---

**Algorithm 1** Process of GSRec.

---

**Input:** historical behavior information of users $S$, the count of users $n$, the count of items $m$, epoch count $T$, learning rate $lr$, embedding size $d$, batch size $B$, GCN layer $L$
**Output:** predict scores $S_{ui}$

---

1: initialize embedding matrix **E**. initialize reverse-order graph $\mathbf{L_r}$ according to Equations (6) and (7).
2: **for** $t \leftarrow 1$ *to* $T$ **do**
3:     take $B$ samples $(u, i_1, i_2, \cdots i_{s_l}, predict\ item, label)$
4:     **for** *layer* $\leftarrow 1$ *to* $L$ **do**
5:         update $E$ according to Equation (14)
6:     **end for**
7:     **for** $b \leftarrow 1$ *to* $B$ **do**
8:         **if** $n > m$ **then**
9:             training sequence is $(u, i_1, i_2, \cdots i_{s_l}, predict\ item)$
10:         **else**
11:             training sequence is $(i_1, i_2, \cdots i_{s_l}, predict\ item)$
12:         **end if**
13:         uses sequence encoder updating **E**
14:         gets predict scores $S_{ui}$ according to equation (22)
15:         gets loss according to equation (23)
16:     **end for**
17:     uses Adam to optimize the Model.
18: **end for**

---

## 5. Experiment

In this section, we perform extensive experiments to evaluate the performance of our model on six real-world datasets and compare our model with different types of current state-of-the-art (SOTA) models. Our experiments are designed to answer the following research questions (RQs):

- **RQ1:** How does GSRec perform compared to the SOTA recommendation methods?
- **RQ2:** Do the reverse-order graph and user embedding affect the performance of GSRec?
- **RQ3:** How do different hyper-parameter settings, such as the dimension of embeddings, the depth of GCN layers, and the number of Transformer blocks, affect GSRec?

### 5.1. Datasets

In our experiments, six common datasets are used to evaluate our model. Three datasets are from Amazon, AmazonBaby (Baby), Amazon Video (Video), and AmazonBook (Book), and the other datasets are ML_100K (100K), Delicious, and LastFM. These datasets are widely used in the research and experimentation of recommendation systems to evaluate the performance and effectiveness of various recommendation algorithms. Among

them, Delicious does not have any sequential signals, which means we have no way of knowing users' sequential behaviors. Here are the descriptions of the datasets:

**100K:** The dataset is a public dataset used for recommendation systems, containing 100,000 movie rating data. Each data row contains information such as userId, movieId, rating, timestamp, movieName, age, gender, occupation, city, etc. In this paper, we only used userId, movieId, and timestamp data. https://grouplens.org/datasets/movielens/ (accessed on 2 January 2024).

**Video:** The dataset is a public dataset that contains a large number of user ratings and comments on video games. This dataset is very useful for studying the performance of recommendation systems in the field of video games. Each data row represents a user's rating and comment on a video game. The fields usually include userId, videoId, rating, timestamps, comment, etc. In this paper, we only used userId, videoId, and timestamps. https://cseweb.ucsd.edu/~jmcauley/datasets/amazon_v2/ (accessed on 2 January 2024).

**Book:** The dataset is a public recommendation system dataset that contains information such as user ratings and comments on books on the Amazon website. Each data row includes userId, bookId, rating, timestamp, userComment, bookTitle, author, etc. In this paper, we only used userId, bookId, and timestamp data. https://cseweb.ucsd.edu/~jmcauley/datasets/amazon_v2/ (accessed on 2 January 2024).

**Baby:** The dataset is a public dataset used for recommendation systems, which includes purchase records and user information of maternal and child products on the Amazon website. Each data row includes userId, productId, timestamp, purchaseQuantity, itemName, itemPrice, and itemtRating, etc. In this paper, we only used userId, movieId, and timestamp data. https://cseweb.ucsd.edu/~jmcauley/datasets/amazon_v2/ (accessed on 2 January 2024).

**Delicious:** The dataset is a publicly available recommendation system dataset that includes user tag records for web pages on the Delicious website. Each data row includes userId, webURL, tag, userPersonalWebPageURL, userFollowedUserList, etc. In this paper, we only used userId and webURL. This dataset does not have any sequential signals. https://grouplens.org/datasets/hetrec-2011/ (accessed on 2 January 2024).

**LastFm:** The dataset is a publicly available recommendation system dataset that includes user listening records and music tags on the Last.fm music recommendation website. Each data row includes userId, artistI, timestamp, listeningFrequency, artistName, artistType, etc. In this paper, we only used userId, artistId, and timestamp. http://www.dtic.upf.edu/ocelma/MusicRecommendationDataset/index.html (accessed on 2 January 2024).

We construct the behavioral sequences of users based on the interaction time between users and items. For the GCN part, we divide the historical behavioral sequences of users into three parts:

- The last item of the behavioral sequence of the user is used as the test set.
- The penultimate item of the behavioral sequence of the user is used as the validation set.
- Other behavioral information of the user is used as the training set. The training set consists of triples (user, positive item, negative item), and the adjacency matrix is constructed according to the training set.

For the sequential part, we also divide the historical behavioral data of users into three parts according to the above rules. The characteristics of these datasets are shown in Table 2. UsersLen represents the average number of interactions a user has with an item (same as ItemsLen).

### 5.2. Evaluation Metric

To compare the recommendation effect of all models, we use two evaluation metrics, namely, Hit Ratio (HIT) and Normalized Discounted Cumulative Gain (NDCG). HR measures how many candidate items are ranked with the TOP-N list, while NDCG accounts for the position of the hit by assigning a higher score to hit at top positions. In this work, we report HR and NDCG with k = 5, 10, or 20. It is essential to point out that each user has

only one ground truth item. Higher values are associated with better model performance for all of these metrics.

$$HIT@K = \frac{1}{N} \sum_{i=1}^{N} Hit(i) \qquad (25)$$

where $N$ is the total number of users and $Hit(i)$ represents whether the item visited by the $i$-th user is in the top K of the recommendation list: 1 if yes, 0 otherwise.

$$NDCG@K = \frac{1}{N} \sum_{i=1}^{N} \frac{1}{log_2(p_i + 1)} \qquad (26)$$

where $N$ is the total number of users and $p_i$ represents the position of the item visited by the ith user in the top K of the recommendation list.

**Table 2.** Statistics of datasets used in this paper. Density means the density of the dataset.

| Dataset | Users | Items | Interactions | UsersLen | ItemsLen | Density |
|---------|-------|-------|--------------|----------|----------|---------|
| 100K | 943 | 1682 | 100,000 | 106.04 | 59.45 | 6.30% |
| Video | 5066 | 1655 | 36,635 | 7.23 | 22.14 | 0.44% |
| Book | 15,041 | 7329 | 182,504 | 12.13 | 24.90 | 0.17% |
| Baby | 19,369 | 6997 | 160,155 | 8.27 | 22.89 | 0.12% |
| Delicious | 1862 | 1862 | 15,329 | 8.23 | 8.23 | 0.44% |
| LastFM | 1893 | 12,524 | 186,470 | 98.51 | 14.89 | 0.79% |

*5.3. Baselines*

To verify the performance of the GSRec model proposed in this paper, we compared the model with the following state-of-the-art recommendation methods. It is mainly divided into three models. The first is collaborative filtering-based (CF-Based), the second is sequence-based, and the last is hybrid-based.

**Collaborative Filtering-Based (CF-Based) Model**

- **NCF [50]**: A modification of MF by replacing the inner product in MF with an MLP.
- **NGCF [23]**: Integrates the user–item bipartite graph structure into the embedding propagation process, enabling expressive modeling of higher-order connectivities in the graph.
- **LightGCN [24]**: An improved version of NGCF that removes the feature transformation and nonlinear activation modules in NGCF. It makes GCN-based methods more concise and suitable for recommendation and achieves SOTA performance.

**Sequence-Based Model**

- **Caser [12]**: Uses a CNN model to capture high-order Markov chains by applying convolutional operations on the embedding of the L recent items, which makes it good at capturing the short-term interests of users.
- **SASRec [15]**: The first to introduce Transformer into recommendation systems. This model uses Transformer to capture the sequential behaviors of users and achieves SOTA performance on sequential recommendations.
- **ComiRec-SA [17]**: Replaced ComiRec-DR's dynamic routing module with a self-attention mechanism.
- **FMLP [51]**: Applied a filter-enhanced all-MLP architecture to the sequential recommendation task.

**Hybrid-Based Model**

- **GES [43]**: Combines GCN and SASRec and is the SOTA method in hybrid recommendations.

*5.4. Parameter Setting*

In order to make sure the comparison is fair, we set the same parameters in different models. The embedding size is 64 and the learning rate is 0.001. The batch size is 4096. The activation function is sigmoid. In sparse datasets, the layer is 2; in dense datasets, the layer is 1. The length is 50; the dropout rate is 0.2 and the L2 regularization is $1 \times 10^{-6}$. In Caser, the number of horizontal filters is 6, the number of vertical filters is 4, and the width of the vertical filter is 1. In transform-based models, the block is 2 and the head is 1. In Comi-DR and Comi-SA, the number of interests is 4 and the route time is 3.

*5.5. Performance Comparison (RQ1)*

We evaluated the performance of all compared methods, and Table 3 shows the results. We can observe that GSRec outperforms different types of baselines on six datasets. This ascertains the effectiveness of our proposed model.

**NCF:** In 100K, its performance surpasses that of GCN-based models because, in 100K, NCF has enough training samples, while GCN-based models suffer from oversmooth problems. The performance of NCF is worse than sequence-based models because, in 100K, the user's behavior sequence is long and sequence-based models can learn the user's sequence information well, while NCF cannot learn the sequential signal. In sparse datasets, the performance of NCF is the worst owing to the fact that training data in sparse datasets is limited and NCF exhibits underfitting. In Delicious, NCF outperforms Caser because Caser is an intense sequence model.

**NGCF:** In 100K, the performance of NGCF only surpassed that of LightGCN, because the GCN-based model experienced smoothing issues in dense datasets. NGCF performs better than LightGCN because its convolutional strategy is more suitable for dense datasets. In Book, Video, Baby, and LastFM, NGCF performs only better than NCF because its convolutional strategy is more suitable for dense datasets.

**LightGCN:** LightGCN performs the worst in 100K and performs better on Book, Video, Baby, and LastFMs. This is because LightGCN's convolution strategy is more suitable for sparse datasets, which can lead to oversmooth problems in dense datasets. In Delicious, the performance of LightGCN is second only to GES, because sequence-based models cannot obtain sequence information and Delicious is relatively sparse, while NGCF and NCF are more suitable for dense datasets.

**Caser:** In 100K, Caser outperformed the GCN-based model and NCF in terms of performance, as the 100K dataset had a great sequential signal, but GCN-based models and NCF did not utilize this sequential signal. In Delicious, Caser performs the worst because the CNN used by Caser extracts sequence information that does not exist.

**SASRec, ComiSA, FMLP:** The above methods have achieved great performance in these datasets because they capture long-term semantic information through self-attention mechanisms and sequence modeling while using relatively few actions for prediction. In addition, in sparse datasets the model can focus on information related to the current prediction, thereby reducing the impact of noise and redundant information. However, the above methods only focus on the user's sequential signal and cannot find spatial signals, so their effectiveness is not as good as that of GES and GSRec.

**GES:** GES achieved good results in the above datasets because GES combines GCN and Transformer. However, due to the attempt of GES to make the short-term behavior of users more significant during GCN, its performance is not as good as FMLP, SASRec, or GSRec in 100K.

**GSRec:** In the above datasets, GSRec achieved the best results, and the results can prove the superiority of GSRec. In addition, we also found that the improvement of NDCG is not as good as HIT, mainly due to GCN. GCN enables the model to discover more items of interest to users, resulting in an improvement in HIT. However due to the increasing number of items that users are interested in, the score difference between each item decreases and the sorting quality of the recommendation list deteriorates to a certain extent, which affects NDCG.

**Table 3.** Performance comparison of all methods on six datasets. The best and the second-best results are highlighted in boldface and underlined, respectively. Imp denotes the improvement of GSRec over the best baseline performer.

| Model | Dataset | HIT@5 | HIT@10 | HIT@20 | NDCG@5 | NDCG@10 | NDCG@20 |
|---|---|---|---|---|---|---|---|
| NCF | 100K | 57.20 | 72.35 | 85.91 | 44.29 | 50.14 | 54.60 |
| | Video | 30.48 | 41.14 | 54.34 | 33.23 | 40.65 | 47.29 |
| | Book | 25.47 | 36.45 | 52.41 | 28.67 | 35.68 | 43.74 |
| | Baby | 15.03 | 23.41 | 36.73 | 22.51 | 30.06 | 39.17 |
| | Delicious | 17.35 | 28.20 | 42.16 | 22.42 | 30.54 | 38.37 |
| | LastFM | 48.81 | 58.90 | 68.89 | 52.03 | 56.65 | 60.72 |
| NGCF | 100K | 55.30 | 71.72 | 85.91 | 43.28 | 49.62 | 54.47 |
| | Video | 30.22 | 41.55 | 55.88 | 31.14 | 38.24 | 45.20 |
| | Book | 35.75 | 52.07 | 67.14 | 36.34 | 43.25 | 49.27 |
| | Baby | 17.34 | 27.12 | 41.28 | 22.21 | 29.89 | 38.78 |
| | Delicious | 39.21 | 46.56 | 56.39 | 49.61 | 53.95 | 58.64 |
| | LastFM | 62.70 | 69.41 | 76.28 | 66.44 | 69.90 | 72.32 |
| LightGCN | 100K | 54.13 | 69.81 | 85.38 | 37.86 | 45.21 | 50.20 |
| | Video | 38.02 | 49.84 | 64.42 | 38.12 | 44.85 | 50.74 |
| | Book | 41.13 | 55.14 | 70.90 | 35.72 | 42.93 | 48.97 |
| | Baby | 20.89 | 30.49 | 43.84 | 26.38 | 33.58 | 41.53 |
| | Delicious | <u>52.63</u> | <u>60.10</u> | <u>67.86</u> | <u>60.12</u> | <u>64.38</u> | <u>67.70</u> |
| | LastFM | 62.18 | 70.89 | <u>77.87</u> | 64.30 | 68.23 | 70.88 |
| Caser | 100K | 59.70 | 72.43 | 86.64 | 44.11 | 50.12 | 54.84 |
| | Video | 34.23 | 45.65 | 60.32 | 41.10 | 43.03 | 47.36 |
| | Book | 47.69 | 61.74 | 75.79 | 38.98 | 45.75 | 50.94 |
| | Baby | 19.95 | 30.42 | 46.411 | 23.46 | 31.35 | 39.85 |
| | Delicious | 10.21 | 16.65 | 25.35 | 19.88 | 29.62 | 37.56 |
| | LastFM | 61.68 | 68.54 | 76.05 | 62.88 | 66.25 | 69.05 |
| SASRec | 100K | 61.28 | 75.19 | <u>88.62</u> | <u>46.70</u> | <u>52.61</u> | 56.73 |
| | Video | <u>40.47</u> | 50.76 | 63.12 | 41.90 | 47.83 | 53.30 |
| | Book | 54.17 | 65.53 | 77.02 | 48.24 | 53.87 | 57.99 |
| | Baby | 21.84 | 31.56 | 45.30 | 27.62 | 35.03 | 42.77 |
| | Delicious | 40.53 | 49.06 | 58.71 | 47.25 | 52.55 | 56.74 |
| | LastFM | 68.65 | 72.73 | 78.43 | 76.80 | 81.90 | 84.01 |
| ComiSA | 100K | 57.90 | 71.76 | 84.09 | 45.60 | 51.23 | 55.45 |
| | Video | 31.85 | 40.02 | 52.06 | 41.64 | 47.08 | 53.07 |
| | Book | 42.14 | 52.77 | 65.24 | 43.27 | 49.19 | 54.17 |
| | Baby | 17.81 | 26.47 | 39.64 | 24.57 | 31.94 | 40.31 |
| | Delicious | 38.88 | 49.47 | 56.41 | 47.36 | 51.84 | 56.64 |
| | LastFM | 61.41 | 67.78 | 73.19 | 79.69 | 81.95 | 83.81 |
| GES | 100K | 60.13 | 74.23 | 86.74 | 46.27 | 52.45 | 56.41 |
| | Video | 40.26 | <u>51.81</u> | 63.67 | 40.27 | 46.71 | 51.86 |
| | Book | <u>57.44</u> | <u>68.19</u> | <u>78.12</u> | <u>49.95</u> | <u>55.36</u> | <u>58.31</u> |
| | Baby | <u>23.83</u> | 32.88 | 44.69 | <u>31.60</u> | <u>38.69</u> | <u>45.55</u> |
| | Delicious | 49.53 | 56.76 | 66.59 | 54.45 | 58.34 | 62.38 |
| | LastFM | 68.81 | <u>73.73</u> | <u>77.62</u> | <u>80.73</u> | <u>81.97</u> | 82.93 |
| FMLP | 100K | <u>61.94</u> | <u>75.26</u> | 88.05 | 46.31 | 52.34 | <u>57.11</u> |
| | Video | 40.90 | 51.64 | 63.96 | **43.13** | **50.02** | **54.76** |
| | Book | 53.82 | 65.13 | 76.65 | 48.59 | 51.13 | 58.18 |
| | Baby | 22.59 | <u>33.32</u> | <u>46.64</u> | 30.02 | 38.67 | 44.44 |
| | Delicious | 43.50 | 52.38 | 60.86 | 50.28 | 55.51 | 59.32 |
| | LastFM | <u>69.28</u> | 73.93 | 77.86 | 80.09 | 81.58 | 84.01 |

**Table 3.** *Cont.*

| Model | Dataset | HIT@5 | HIT@10 | HIT@20 | NDCG@5 | NDCG@10 | NDCG@20 |
|---|---|---|---|---|---|---|---|
| GSRec | 100K | **64.58** | **78.69** | **91.73** | **49.72** | **55.23** | 59.32 |
| | Video | **45.05** | **57.16** | **69.18** | <u>42.85</u> | <u>49.85</u> | <u>54.50</u> |
| | Book | **62.87** | **74.08** | **84.34** | **52.50** | **57.51** | 60.84 |
| | Baby | **27.15** | **38.08** | **52.07** | **31.79** | **38.76** | 45.60 |
| | Delicious | **55.71** | **62.59** | **69.82** | **60.76** | **64.35** | 67.41 |
| | LastFM | **75.35** | **80.11** | **84.59** | **81.18** | **83.39** | 85.91 |
| Imp | 100K | 4.26% | 4.60% | 3.51% | 6.52% | 5.01% | 3.87% |
| | Video | 11.90% | 10.33% | 8.37% | -0.65% | -0.34% | -0.47% |
| | Book | 9.45% | 8.64% | 7.96% | 5.11% | 3.88% | 4.34% |
| | Baby | 13.93% | 14.29% | 11.67% | 0.50% | 0.18% | 0.11% |
| | Delicious | 5.85% | 4.14% | 2.93% | 1.06% | 0.11% | 0.03% |
| | LastFM | 8.69% | 7.92% | 7.81% | 1.07% | 1.76% | 1.74% |

*5.6. Ablation Study (RQ2)*

In this section, we answer question 2: Do the reverse-order graph and the user embedding affect GSRec? At first, we tried to construct a forward-order graph and add a penalty factor to items purchased or clicked far back in the adjacency matrix.

$$\begin{cases} a(u_i, i_j) = 0.5 + (1 - \theta \times index)/2 & u_i \text{ interact } i_j \\ a(u_i, i_j) = 0 & \text{other} \end{cases} \tag{27}$$

This method is effective in GCN-based models but pays more attention to the recent behaviors of users, and the effect in GSRec is worse than without the forward-order graph. This probably places more emphasis on recent behaviors than on holistic behaviors. We then construct the reverse-order graph and add an incentive factor to items purchased or clicked far back in the adjacency matrix. The results are shown in Table 4, and we can see that the effect in GSRec is better than without the reverse-order graph. In these datasets, the incentive factor improves the effect of the model, which means that it is necessary to use the reverse-order graph in sequential and sparse datasets.

**Table 4.** The effect of incentive factor; GSRec-R means the model uses the reverse-order graph and GSRec-NO-R means the model does not use the reverse-order graph.

| Dataset | Model | HIT@5 | HIT@10 | HIT@20 | NDCG@5 | NDCG@10 | NDCG@20 |
|---|---|---|---|---|---|---|---|
| Video | GSRec-NO-R | 43.79 | 55.02 | 67.11 | 42.68 | 48.90 | 54.07 |
| | GSRec-R | 45.05 | 57.16 | 69.18 | 42.85 | 49.85 | 54.50 |
| | Imp | 2.88% | 3.89% | 4.02% | 0.40% | 2.00% | 0.80% |
| Book | GSRec-NO-R | 62.70 | 73.89 | 84.18 | 52.12 | 57.29 | 60.64 |
| | GSRec-R | 62.87 | 74.08 | 84.34 | 52.50 | 57.51 | 60.84 |
| | Imp | 0.27% | 0.26% | 0.19% | 0.73% | 0.38% | 0.33% |
| Baby | GSRec-NO-R | 26.93 | 37.08 | 51.00 | 30.76 | 38.69 | 45.55 |
| | GSRec-R | 27.15 | 38.08 | 52.07 | 31.79 | 38.76 | 45.6 |
| | Imp | 0.82% | 2.70% | 2.10% | 3.34% | 1.84% | 1.15% |
| LastFM | GSRec-NO-R | 73.57 | 78.38 | 82.76 | 79.05 | 81.17 | 82.93 |
| | GSRec-R | 74.11 | 78.81 | 83.35 | 81.18 | 83.28 | 84.79 |
| | Imp | 0.73% | 0.55% | 0.71% | 2.69% | 2.60% | 2.24% |

To investigate whether user embedding works in Transformer, we compare GSRec-NO-USER and GSRec. Table 5 summarizes the experimental results. This table shows that user embedding benefits our model in datasets where the number of users is more than the number of items.

**Table 5.** The effect of the user embeddings; GSRec-U means the model uses the user embeddings and GSRec-NO-U means the model does not use the user embeddings.

| Dataset | Model | HIT@5 | HIT@10 | HIT@20 | NDCG@5 | NDCG@10 | NDCG@20 |
|---------|-------|-------|--------|--------|--------|---------|---------|
| 100K | GSRec-NO-U | 63.20 | 76.88 | 89.61 | 47.51 | 53.60 | 57.43 |
| | GSRec-U | 64.58 | 78.69 | 91.73 | 49.72 | 55.23 | 59.32 |
| | Imp | 2.18% | 2.35% | 2.37% | 4.65% | 3.04% | 3.29% |
| Delicious | GSRec-NO-U | 55.29 | 62.21 | 69.53 | 59.78 | 63.32 | 66.17 |
| | GSRec-U | 55.71 | 62.59 | 69.82 | 60.76 | 64.35 | 67.41 |
| | Imp | 0.76% | 0.61% | 0.42% | 1.64% | 1.63% | 1.87% |
| LastFM | GSRec-NO-U | 73.57 | 78.38 | 82.76 | 79.05 | 81.17 | 82.93 |
| | GSRec-U | 75.30 | 79.57 | 83.95 | 80.09 | 82.17 | 84.76 |
| | Imp | 2.35% | 1.52% | 1.44% | 1.32% | 1.23% | 2.21% |

### *5.7. Parameter Sensitivity Analysis (RQ3)*

In this section, we answer question 3: How do different hyper-parameter settings affect GSRec? In order to answer RQ3, we mainly analyze the dimension of the embeddings, the depth of GCN layers, and the number of Transformer blocks. We use the Sum metrics.

$$Sum = HIT@5 + HIT@10 + HIT@20 + NDCG@5 + NDCG@10 + NDCG@20 \quad (28)$$

**Influence of the embedding dimension:** As can be seen from Figure 3, the embedding dimension is tuned from [8, 16, 32, 64, 128]. We can find that the change in the embedding dimension has a specific influence on the quality of the model. When the value is 8, the result is the worst; this may be because the embedded dimensions are too small, resulting in the model only learning some of the features from the dataset. The results are better when gradually increasing to 64, and when the value is 128 the effect in some datasets is worse than 64. When the embedding dimension is too large, the model becomes too complex and overfitting may occur. On the other hand, as the embedding dimension increases, the number of model parameters also increases, making it hard to optimize the model. In addition, if the embedding dimension is too large, the model may capture more noise and irrelevant information. Therefore, we believe that the embedding dimension has a specific impact on the model. With the increase of the embedding dimension, the training time also increases, so we choose the embedding size of 64.

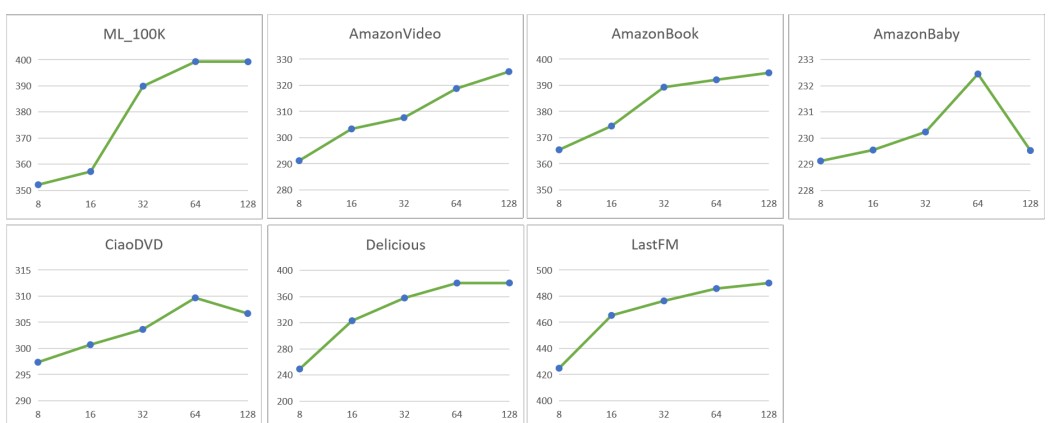

**Figure 3.** Influence of the embedding dimension.

**Influence of the number of GCN layers:** To research whether the number of GCN layers is helpful for our model, we change the count of layers, which searches the layer numbers in the range of [1, 2, 3, 4]. Figure 4 summarizes the experimental results. Through experiments, we have the following observations. In dense datasets, the model works best when the layer is 1, and in sparse datasets the model works best when the layer is 2 or 3.

Therefore, the higher the number of layers, the worse the effect of the model. This may be because too many GCN layers can lead to signal convergence and loss of diversity in node features, which may have a negative impact on the learning performance of the model. All in all, we can see that the number of layers has an effect, even though the effect is relatively small.

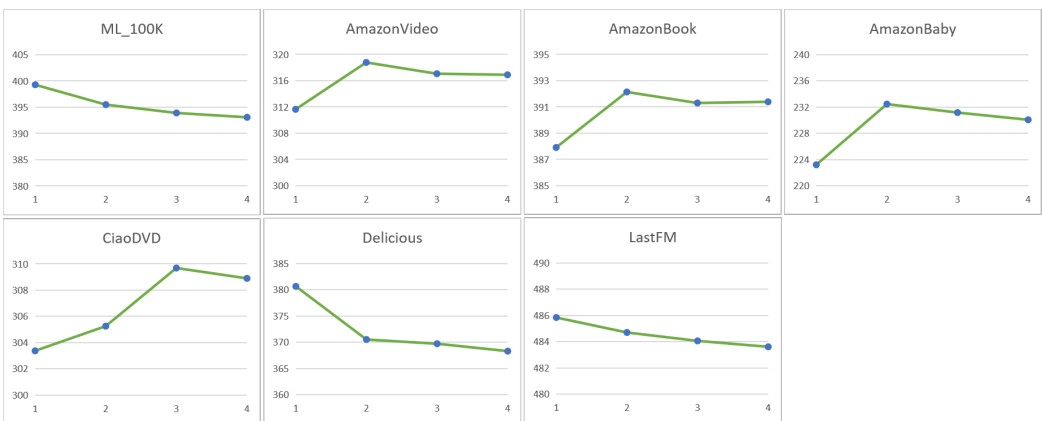

**Figure 4.** Influence of the depth of the GCN layer.

**Influence of the number of Transformer blocks:** To research whether the number of blocks is helpful for our model, we change the count of blocks, which searches the block numbers in the range of [1, 2, 3]. Figure 5 summarizes the experimental results. Through the experiments, we obtain the following results. As the number of blocks increases, the model's performance will improve at the beginning. When the number of blocks is 1, the model can only capture partial features of the data. When the number of blocks is 2, the model's performance is optimal because the depth of the model is moderate and the number of parameters of the model is also appropriate, which can capture different features of input data without being too complex. When the number of blocks is 3, the performance of the GSRec degrades because the number of model parameters increases and overfitting occurs.

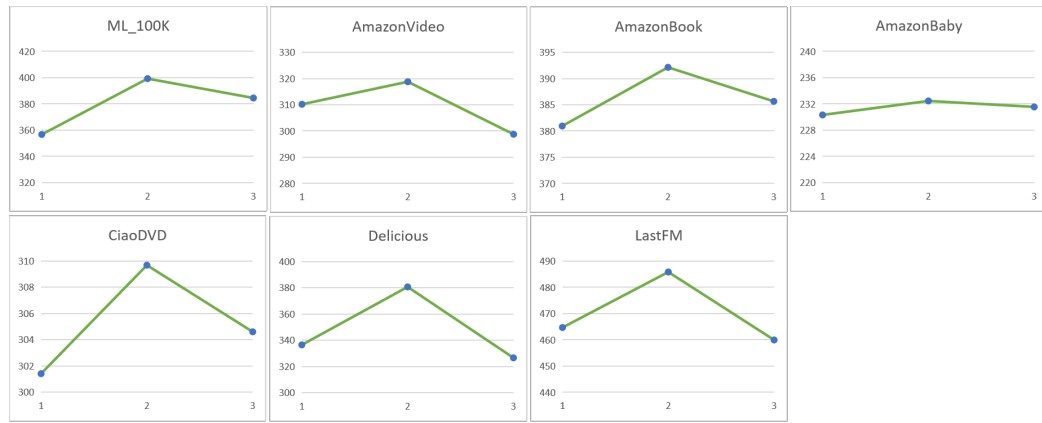

**Figure 5.** Influence of the number of Transformer blocks.

## 6. Discussion

GSRec achieved SOTA performance on six datasets, demonstrating the superiority of GSRec. However, Table 6 shows that GSRec also has some drawbacks, especially in computational efficiency. Caser, SASRec, ComiSA, GES, and FMLP only used item embeddings and did not use user embeddings. Therefore, we can see that their models have fewer parameters and higher computational efficiency, which is particularly evident in datasets with fewer users. LightGCN and NGCF have a large number of parameters and poor computational efficiency due to the use of user embeddings. NCF has two user

embeddings and two item embeddings; thus, compared to other models, NCF has the highest number of parameters and the lowest computational efficiency. GSRec has a large number of parameters and computational complexity, resulting in low computational efficiency. This is because GSRec not only uses user and item embeddings, but also uses Transformers. This leads to its lower real-time performance compared to sequence-based models or collaborative filtering-based models, making it difficult for it to play a role in scenarios with high real-time requirements. However, we believe that this difference is not significant, so it is acceptable in the vast majority of scenarios. In addition, GSRec uses GCN, which requires recalculating the matrix values when new users enter. In recommendation systems, the number of model parameters is largely determined by the embeddings and the embeddings are determined by the number of users and items. Therefore, the more users and items there are, the larger the parameters of the model. How to design a lightweight and efficient hybrid recommendation model is our future research direction.

**Table 6.** Time consumption comparison of all methods on six datasets. The metric is the number of parameters in the model (Mb).

| Model | 100K | Video | Book | Baby | Delicious | LastFM |
|-------|------|-------|------|------|-----------|--------|
| NCF | 0.87 | 2.15 | 7.25 | 7.94 | 1.24 | 4.36 |
| NGCF | 0.68 | 1.65 | 5.47 | 6.46 | 0.92 | 3.54 |
| LightGCN | 0.64 | 1.64 | 5.46 | 6.44 | 0.91 | 3.52 |
| Caser | 0.48 | 0.47 | 1.86 | 1.78 | 0.52 | 3.13 |
| SASRec | 0.57 | 0.56 | 1.95 | 1.87 | 0.62 | 3.22 |
| ComiSA | 0.59 | 0.58 | 2.02 | 1.93 | 0.67 | 3.31 |
| GES | 0.73 | 0.72 | 1.98 | 1.91 | 0.77 | 3.27 |
| FMLP | 0.52 | 0.52 | 1.91 | 1.84 | 0.61 | 3.12 |
| GSRec | 0.82 | 1.66 | 5.48 | 6.46 | 1.07 | 3.54 |

As for the cold start problem, the focus of GSRec is to combine the long-term and short-term interests of users. GSRec does not focus on the cold start problem in recommendation systems, but compared to sequential recommendation models GSRec can alleviate the cold start problem to some extent. When a new user enters, GSRec cannot understand their interests as they have not purchased or browsed related items. A usual method for this is to recommend popular items. When a user purchases an item, GSRec calculates the similarity between the new user and the user who has purchased the item through GCN and predicts which items the new user may like based on the similarity. In summary, GSRec is not meant to solve the cold start problem, but rather to address the combination of long-term and short-term interests. The commonly used methods for cold start problems include providing non-personalized recommendations, utilizing information provided by new users during registration and content-based recommendations, and by adopting quick probing strategies.

Since the dataset we are using has excluded users with behavior of less than 4, we will recommend users with behavior equal to 4 to evaluate whether the model can have good recommendation performance under cold start conditions. Due to the relatively dense nature of the 100K and LastFM datasets, we chose to use Video, Baby, Book, and Delicious as the test datasets. The baselines we selected were LightGCN, SASRec, GES, and Caser. It should be emphasized that these baselines and GSRec are not designed to solve the cold start problem. We also do not believe that the effectiveness of GSRec in cold start scenarios can exceed that of models specifically designed to solve cold start problems. From Table 7, we can observe that LightGCN achieved good results in the cold start scenario. Therefore, compared to the sequence model LightGCN, it can aggregate the information of neighboring nodes through convolution. The performance of sequence models deteriorates in cold start scenarios because they can only obtain the user's data and cannot perform data augmentation. GSRec achieved the best performance in all four datasets because it enhanced user embedding information and item embedding information by aggregating

neighboring nodes using GCN, making the embeddings used by Transformer contain more information.

**Table 7.** Performance comparison of different methods on four datasets. The best results are highlighted in boldface and the second-best results are highlighted underlined, respectively. Imp denotes the improvement of GSRec over the best baseline performer.

| DataSet | Model | HIT@5 | HIT@10 | HIT@20 | NDCG@5 | NDCG@10 | NDCG@20 |
|---------|-------|-------|--------|--------|--------|---------|---------|
| Video | LightGCN | 41.02 | 52.03 | 64.24 | 41.49 | 48.16 | 53.38 |
| | Caser | 38.21 | 50.84 | 65.25 | 37.20 | 44.17 | 50.18 |
| | SASRec | 37.67 | 48.88 | 63.09 | 38.25 | 44.28 | 50.12 |
| | GES | 39.24 | 51.82 | 64.17 | 41.06 | 47.03 | 52.14 |
| | GSRec | **44.12** | **56.34** | **67.14** | **43.21** | **52.62** | **55.77** |
| | Imp | 7.55% | 8.23% | 4.51% | 4.15% | 9.26% | 4.48% |
| Book | LightGCN | 49.44 | 60.90 | 73.55 | 45.84 | 51.32 | 55.98 |
| | Caser | 47.61 | 60.95 | 75.09 | 39.75 | 46.35 | 51.41 |
| | SASRec | 48.05 | 59.42 | 73.63 | 42.99 | 48.76 | 54.11 |
| | GES | 50.44 | 61.53 | 75.32 | 44.35 | 49.60 | 56.37 |
| | GSRec | **52.84** | **63.71** | **78.17** | **48.97** | **54.40** | **57.63** |
| | Imp | 4.76% | 3.54% | 3.78% | 6.83% | 6.00% | 2.95% |
| Baby | LightGCN | 19.33 | 29.00 | 41.87 | 25.52 | 33.23 | 41.04 |
| | Caser | 20.25 | 29.64 | 44.13 | 24.20 | 32.28 | 40.56 |
| | SASRec | 21.04 | 30.74 | 45.72 | 25.7 | 33.24 | 41.47 |
| | GES | 22.80 | 31.39 | 44.98 | 28.95 | 35.17 | 43.44 |
| | GSRec | **25.82** | **36.21** | **49.55** | **30.02** | **36.12** | **44.30** |
| | Imp | 13.24% | 15.36% | 8.38% | 3.59% | 2.70% | 1.98% |
| Delicious | LightGCN | 16.77 | 28.31 | 42.84 | 24.64 | 34.05 | 42.82 |
| | Caser | 7.38 | 15.09 | 28.57 | 11.45 | 20.74 | 30.2 |
| | SASRec | 12.36 | 21.03 | 37.24 | 19.22 | 27.5 | 37.21 |
| | GES | 14.17 | 25.57 | 40.06 | 22.56 | 30.47 | 39.98 |
| | GSRec | **19.37** | **31.46** | **44.29** | **27.57** | **37.73** | **43.08** |
| | Imp | 15.50% | 11.12% | 3.38% | 11.90% | 10.80 | 0.61% |

## 7. Conclusions

In this work, we propose a novel framework for sequence-based recommendation, called GSRec. In GSRec, we construct a user–item reverse-order graph from the interaction between users and items, and we use an incentive factor to make the model more focused on the long-term behaviors of users. In addition, we try to capture the transfer information for the sequence of items. Transformer is developed to encode sequential signals. Finally, learned item embedding is used to make the final recommendation. We conduct experiments on six existing datasets showing that our method is more advanced than the current SOTA methods.

**Author Contributions:** Data curation, X.M.; formal analysis, X.M. and X.Y.; methodology, X.M., L.Z. and X.K.; resources, J.T.; supervision, X.K.; validation, J.T.; writing—original draft, X.M. and X.K.; writing—review and editing, J.T., L.Z. and X.Y. All authors have read and agreed to the published version of the manuscript.

**Funding:** This research was funded in part by the National Natural Science Foundation of China under Grant 62072409 and Grant 62176234, in part by Key Research Project of ZheJiang Lab under Grant 2022NF0AC01, in part by the Zhejiang Provincial Natural Science Foundation under Grant LR21F020003, and in part by the Fundamental Research Funds for the Provincial Universities of Zhejiang under Grant RF-B2020001.

**Data Availability Statement:** This research employed publicly available datasets for its experimental studies.

**Conflicts of Interest:** The authors declare no conflicts of interest.

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
