# Peer review of "GSRec: A Graph-Sequence Recommendation System Based on Reverse-Order Graph and User Embedding"

_mathematics, doi:10.3390/math12010164_

Round 1
Reviewer 1 Report
Comments and Suggestions for Authors
The paper presents GSRec, a graph-sequence recommendation system that integrates Graph Convolution Network (GCN) and Transformer. It addresses limitations of sequence-based models in recommendations by considering both sequential and spatial data signals. GSRec employs a reverse-order graph in GCN and user embedding in Transformer, improving efficiency and accuracy. Tests on six datasets.
Comments
1. Clarification on the choice of datasets and their relevance to the study would strengthen the paper.
2. More detailed discussion on the limitations of your approach would provide a balanced view.
3. Exploring the impact of various hyperparameters in greater depth could be insightful.
4. Elaborate on how your model handles new users or items (cold start problem).
5. Discuss the scalability of your model in larger, more complex datasets.
6. Clarify the computational efficiency of your model compared to existing systems.
7. How does GSRec handle sparse data scenarios?
8. Can you elaborate on the model's performance in cold start situations?
9. Did you explore the impact of different hyperparameters on model performance?
10. How does GSRec's computational efficiency compare with other models?
11. Can GSRec be adapted for different types of recommendation scenarios beyond products?
Reviewer 2 Report
Comments and Suggestions for Authors
Authors proposed a graph sequence-based model called GSRec that combines 8 Graph Convolution Network (GCN) and Transformer. Experimental results demonstrate significantly improved performance on benchmark datasets. In addition, the follow suggestions maybe further improve the manuscript:
1)The motivation of this manuscript is not clear enough. What problem does the article solve, and what is the value of this problem?
2) This manuscript utilizes an off-the-shelf transformer for user embedding, and a combination of self-attention mechanisms and reverse-order graph to capture temporal information. These technical solutions are merely an amalgamation of existing tools and lack novelty.
3) The manuscript provides only a cursory introduction to the use of the loss function (Binary Cross Entropy). Please introduce the manuscript's loss function and the optimization process using formal expressions.
4) The authors did not well summarize the challenges or problems appearing in the previous studies. Even though they introduce some related work in the first section, they still should introduce what disadvantages that they have.
5) The paper lacks the deep discussion of the experimental results. For example, why does the proposed approach perform well. The authors should introduce the limitations of the proposed approaches.
6) The use study should be added to describe the evidences to demonstrate the effectiveness of the proposed approach in the real world.
Overall, This paper is well written and presents an important method to implement Graph Neural Network.
Reviewer 3 Report
Comments and Suggestions for Authors
Review of "GSRec: A Graph-Sequence Recommendation System Based on Reverse-Order Graph and User Embedding"
The authors propose a new graph sequence-based model called GSRec. The methodology is clearly written with discussions. Experiments are performed to show the performance of the proposed method via comparison study with appropriate real datasets and evaluation metrics. I believe this work may have a great impact on the recommendation system in user-item structure and be widely read.
The following are minor points:
Minor Comments
Line 59: Relate -> Related
Line 75: archived -> achieved
Line 81: bert -> Bidirectional Encoder Representations from Transformers (BERT)
Line 91: Because of deep learning -> With the rising of deep learning methods
Line 129: The definition of identity matrix I_N is vague. What is N? I assume it is n (the number of users).
Table 1: count -> number, set -> the set
Line 145: Figure.1 -> Figure 1, please check others too.
Line 152, 153: a set of -> the set of
Eq. (7): Please define "index."
Eq. (8): Please clarify "N" of I_N.
Eq. (10): Please check the position of (k+1) and (k). I assume they should be superscripts of e.
Line 195: Please explain why the authors set the identical dimensions for all the layers. The last layer would be a good candidate; in this case, the layers do not necessarily have a different number of units.
Line 200: convolution -> convolutional
Line 207: What if the number of users equals the number of items?
Line 213: Could the authors comment on why random number padding is used?
Eq. (15): Subscripts do not match with Eq. (5).
Line 232: Biases are also learnable parameters.
Line 240: Authors should check the font of x and define x clearly.
Eq. (22): Authors should write the definition of N_i^T.
Table 2: Please clarify what "Dense" means.
Round 2
Reviewer 1 Report
Comments and Suggestions for Authors
The authors addressed my concerns.
Reviewer 2 Report
Comments and Suggestions for Authors
Accept